# Lignin induced iron reduction by novel sp., *Tolumonas lignolytic* BRL6-1

Gina Chaput[1], Andrew F. Billings[1], Lani DeDiego[1], Roberto Orellana[2], Joshua N. Adkins[3], Carrie D. Nicora[3], Young-Mo Kim[3], Rosalie Chu[4], Blake Simmons[5], Kristen M. DeAngelis[1]*

1 Department of Microbiology, University of Massachusetts–Amherst, Amherst, MA, United States of America, 2 Departamento de Biología, Facultad de Ciencias Naturales y Exactas, Universidad de Playa Ancha, Playa Ancha, Valparaíso, Chile, 3 Biological Sciences Department, Pacific Northwest National Laboratory, Richland, Washington, DC, United States of America, 4 Environmental and Molecular Sciences Laboratory, Pacific Northwest National Laboratory, Richland, Washington, DC, United States of America, 5 U.S. Department of Energy Joint Genome Institute, Berkeley, California, United States of America

* deangelis@microbio.umass.edu

**Data Availability Statement:** All relevant data are within the manuscript and its Supporting Information files.

**Funding:** B.S.; K.M.D.; This work was part of the DOE Joint BioEnergy Institute (http://www.jbei.org)

## Abstract

Lignin is the second most abundant carbon polymer on earth and despite having more fuel value than cellulose, it currently is considered a waste byproduct in many industrial lignocellulose applications. Valorization of lignin relies on effective and green methods of de-lignification, with a growing interest in the use of microbes. Here we investigate the physiology and molecular response of the novel facultative anaerobic bacterium, *Tolumonas lignolytica* BRL6-1, to lignin under anoxic conditions. Physiological and biochemical changes were compared between cells grown anaerobically in either lignin-amended or unamended conditions. In the presence of lignin, BRL6-1 accumulates higher biomass and has a shorter lag phase compared to unamended conditions, and 14% of the proteins determined to be significantly higher in abundance by $\log_2$ fold-change of 2 or greater were related to Fe(II) transport in late logarithmic phase. Ferrozine assays of the supernatant confirmed that Fe(III) was bound to lignin and reduced to Fe(II) only in the presence of BRL6-1, suggesting redox activity by the cells. LC-MS/MS analysis of the secretome showed an extra band at 20 kDa in lignin-amended conditions. Protein sequencing of this band identified a protein of unknown function with homology to enzymes in the radical SAM superfamily. Expression of this protein in lignin-amended conditions suggests its role in radical formation. From our findings, we suggest that BRL6-1 is using a protein in the radical SAM superfamily to interact with the Fe(III) bound to lignin and reducing it to Fe(II) for cellular use, increasing BRL6-1 yield under lignin-amended conditions. This interaction potentially generates organic free radicals and causes a radical cascade which could modify and depolymerize lignin. Further research should clarify the extent to which this mechanism is similar to previously described aerobic chelator-mediated Fenton chemistry or radical producing lignolytic enzymes, such as lignin peroxidases, but under anoxic conditions.

supported by the U. S. Department of Energy, Office of Science, Office of Biological and Environmental Research, through contract DE-AC02-05CH11231 between Lawrence Berkeley National Laboratory and the U. S. Department of Energy. K.M.D.; This research was performed using resources at the DOE Environmental Molecular Sciences Laboratory (grid.436923.9), which is an DOE Office of Science User Facilities sponsored by the Office of Biological and Environmental Research and operated under Contract No. DE-AC05-76RL01830. G.C.; This publication was also developed under Assistance Agreement No. FP-91782301-0 awarded by the U. S. Environmental Protection. The funders had no role in study design, data collection and analysis, decision to publish, or preparation of the manuscript.

**Competing interests:** The authors have declared that no competing interests exist.

## Introduction

The industrial processing of lignocellulosic material produces $5x10^6$ metric tons of lignin annually worldwide [1]. Lignin is the largest renewable source of aromatics that can be used for products such as flavors, fragrances, dyes, and other valuable secondary metabolites [2, 3]. However, it is considered an "untapped" resource due to the recalcitrant nature of the polymer, making it difficult to separate and process for valuable downstream products [4, 5].

Investigation of microbially mediated processes for the depolymerization of lignin have focused predominantly on aerobic fungi and bacteria [4, 6–10]. Under oxic conditions, enzymes such as laccases and peroxidases produce oxidants that diffuse into and reduce the lignin complex [9, 11], causing bond scission reactions between lignin subunits. For microorganisms that lack lignolytic enzymes such as brown-rot fungi as well as aerobic bacteria like *Pantoea ananatis* Sd-1 and *Cupriavidus basilensis* B-8, chelator-mediated Fenton chemistry (CMF) is used to depolymerize lignin [12–15]. In this mechanism, the microorganism produces an iron reducing molecule, a chelator molecule, and $H_2O_2$. Once the chelator binds to Fe(III) in the environment, it then reacts with the iron reducing molecule to reduce Fe(III) to Fe(II). Fe(II) then reacts with $H_2O_2$ to create •OH radicals. Similarly to oxidants formed by laccases and peroxidases, the •OH radicals disrupt the lignin structure, causing bond scissions of subunits [15]. Both lignolytic enzyme and chelator mediated lignin depolymerization are promising for industries that rely lignocellulosic feedstocks [8]. For example, the use of Fenton chemistry for lignocellulosic processing has been studied using the aerobic bacterium, *C. basilensis* B-8 grown on rice straw, showing a synergistic relationship in lignin depolymerization and cellulose yield [14]. However, limitations to these processes hinder them from being competitive on the market. Both aerobic fungi and bacteria require constant aeration and mixing, making it very costly to maintain the cultures [16]. Mass production is not possible due to lacking a method of recycling the enzymes after one use, low substrate specificity, and low redox potential [17]. For example, both laccases and manganese-dependent peroxidases cannot degrade 80–90% of lignin due to the presence of non-phenolic structures [18].

Thirty years ago anaerobic bacteria were thought to not have a role in aromatic compound degradation [19], specifically in the degradation of lignin. However, more recent findings have confirmed they do have these capabilities, making them to be a suitable alternative to aerobic microorganisms for industrial applications. For example, based on genome analysis, *Klebsiella sp*. strain BRL6-2 is hypothesized to use lignin as an electron acceptor for energy production [20]. Support for this mechanism is also seen with humic substances, which are lignin rich compounds, that can act as extracellular electron acceptors for bacteria in sediments and anoxic waters [21, 22]. *Enterobacter lignolyticus* SCF1 has been studied comparing the growth in lignin-amended and unamended conditions under anaerobic conditions. RNAseq analysis suggested various enzymes that may be responsible for lignin depolymerization, including alcohol dehydrogenases [23, 24]. However, the exact mechanism has yet to be elucidated. By identifying additional anaerobic bacteria capable of degrading lignin, mechanisms and their regulation can be uncovered and further utilized for lignin depolymerization and valorization applications.

*Tolumonas lignolytica* BRL6-1 is a novel, facultative anaerobic soil bacterium that was isolated from the El Yunque experimental forest, Puerto Rico, on lignin as the sole carbon (C) source under anoxic conditions [25]. Previously, BRL6-1 was demonstrated to have a shorter lag phase and a higher biomass in the presence of lignin compared to unamended conditions [25]. However, the mechanism of lignin modification and how it benefits cell growth is not well understood. Billings *et al*. had suggested that BRL6-1 may be using lignin as a secondary carbon source as well as energy source. In this study, we address this hypothesis by comparing

the physiology of BRL6-1 in the presence of lignin-amended to unamended conditions. Our differential cellular protein abundance data indicates that BRL6-1 shifts from glycolysis to the Entner–Doudoroff (ED) pathway in the presence of lignin. Metabolomic analysis of these same samples also points towards a higher turnover of tricarboxylic acid cycle (TCA) intermediates in the presence of lignin compared to unamended conditions. Detection of benzoic acid in BRL6-1 biomass at late logarithmic phase in the presence of lignin suggests that lignin may be degrading and monomers are entering the cell; however, genes homologous to lignin catabolic enzymes did not change in abundance. In addition, there was a lower abundance of NADH dehydrogenases present in BRL6-1 when cultured in lignin-amended conditions, suggesting a different mechanism for energy production. Based on these results, we further explored a post hoc hypothesis that when grown anaerobically in the presence of lignin, BRL6-1 secretes a protein that acts as both iron chelator and redox agent. By isolating proteins that were secreted, we detected an extracellular protein that potentially generates organic free radicals and causes a radical cascade that modifies and depolymerizes lignin.

## Materials and methods

### Culturing *Tolumonas lignolytica* BRL6-1

To study lignin modification under anoxic conditions, *Tolumonas lignolytica* BRL6-1 was grown in 0.04% D-glucose as the primary C source amended or unamended with 0.1% alkali lignin, low sulfonate (Sigma Aldrich, CAS Number 8068-05-1). Alkali lignin was chosen for this study due to its high purity, which was previously shown with nuclear magnetic resonance (NMR) and high pressure liquid chromatography HPLC analyses [26]. Cultures grew on modified CCMA media consisting of (per liter) 2.25 g NaCl, 0.5 g $NH_4Cl$, 0.227 g $KH_2PO_4$, 0.348 g $K_2HPO_4$, 5 mg $MgSO_4\bullet7H_2O$, 2.5 mg $CaCl_2\bullet2H_2O$, 0.01 mL SL-10 trace elements, and 0.01 mL Thauer's vitamins [27–29]. The D-glucose concentration was 0.2% for the ferrozine assays, Arnow assays, proteome and secretome analysis, described in more detail below. For each experimental condition, cultures grew at 30˚C anaerobically (n = 3). Uninoculated bottles served as abiotic controls (n = 3, unless stated otherwise). Iron amended cultures had an additional 38 ppb Fe(II) added to the media as $FeCl_2\bullet4H_2O$.

To study the physiological response of BRL6-1 in the presence of lignin, growth in lignin-amended medium, lignin unamended medium, and lignin unamended medium supplemented with the additional Fe (II) were monitored by measuring cell concentration by absorption ($OD_{600}$). Bacterial growth curves were analyzed with gcFit function via *grofit* package in R [30]. Calculated average lag phase, maximum growth rate ($\mu$ Max), and maximum cell growth (A) were based on the Gompertz Model.

### Analysis of cellular proteins, secretome, and metabolites

To identify cellular (cytosolic and membrane bound) proteins differentially expressed during lignin-amended growth, biomass was collected at both late logarithmic and mid-stationary phase from cultures grown in the presence or absence of lignin. Cell pellets were lysed via sonification and transferred to PCT MicroTube barocycler pulse tubes with 150µl caps (Pressure Biosciences Inc., South Easton, MA). The MicroTubes were placed in a MicroTube cartridge and barocycled for 10 cycles (20 seconds at 35,000 psi back down to ambient pressure for 10 seconds). All of the material was removed from the MicroTubes and transferred to 1.5mL micro-centrifuge tubes for MPLex (metabolite, protein, lipid extraction) by adding cold (-20˚C) cholorform:methanol mix (prepared 2:1 (v/v)) in a 5:1 ratio (v/v) over sample volume and vigorously vortexed [31]. The sample was then placed on ice for 5 mins and then vortexed for 10 secs followed by centrifugation at 10,000 *xg* for 10 mins at 4˚C. The upper water-soluble

metabolite phase was collected into a glass vial and dried to complete dryness in a speed vac and then stored at -20°C until analysis. The remaining protein interlayer was washed with 1mL of cold methanol, centrifuged at 10,000 *xg* for 5 mins and the supernatant removed, the samples were then placed in a fume hood to dry. Enhanced Filter Aided Sample Preparation (eFASP) [32, 33] was followed according to the protocol for protein digestion.

Raw mass spectrometry data were searched using MS-GF+ against NCBI RefSeq *Tolumonas* sp. BRL6-1 database in addition to bovine/porcine trypsin and other common contaminants such as keratin sequences (3164 total sequences). Searching parameters required tryptic digestion of at least one of the peptide ends (partially tryptic), <10 ppm peptide mass tolerance and methionine oxidation as variable modification. The identified MS/MS spectra were filtered with an MS-GF+ score of $1e^{-09}$ resulting in $\leq 1.0\%$ false discovery rate (FDR) at the protein level. The count of spectra attributed to each individual protein within each experimental condition was used as a proxy for relative quantitative values.

Supernatant fractions from late logarithmic growth phase were collected to identify differentially expressed proteins in the secretome during lignin-amended growth. Cell-free supernatant was generated by vacuum filtration of spent growth media (100mL) with a 0.45 μm filter. Cell-free supernatant was then ultrafiltrated with a 10 kDa Amicon filter. The >10 kDa fraction was retained and proteins were further concentrated using trichloroacetic acid (TCA) precipitation [41]. Samples were run on a 15% SDS-PAGE gel and silver stained. Bands of interest from both lignin-amended and unamended samples were cut out at 20 kDa, 37 kDa, and 50 kDa. Using an in-gel tryptic digest kit (Thermo Fisher, Catalog #89871), samples were prepared as described by the manufacturer for LC-MS/MS analysis. LC-MS/MS analysis was completed by the Mass Spectrometry Center at University of Massachusetts Amherst. Raw mass spectrometry data was search with MS/GF+ against the NCBI RefSeq *T. lignolytica* BRL6-1 database.

## Statistical analysis of proteomic and metabolomic data

Spectral counts, which represent protein abundance, from the cellular proteomics between lignin-amended and unamended conditions were compared using msms.edgeR function via *msms.Tests* package in R [34]. The post-test effect size filter of msms.edgeR deemed proteins differentially expressed if proteins had p-values <0.05, absolute values of $\log_2$ fold-change >1 or < -1, and total spectral counts >2 across biological replicates [34].

Metabolomic abundances were z-transformed for each compound detected with an additional pseudo count added to have all positive values (lowest value plus 1 per compound). To determine whether metabolite abundances differed based on lignin treatment, growth phase, or both, we performed MANOVA using residual randomization in permutation procedures (RRPP) with the RRPP package in R [35]. Non-metric multidimensional scaling (NMDS) ordination method, calculated on the basis of Bray-Curtis distances, was completed with the vegan package in R [36]. To identify metabolites that were significantly higher in either lignin-amended or unamended conditions across growth phases, pairwise indicator species analysis was completed with the labdsv package in R [37]. Metabolite abundances of indicators with known identification were visualized using the heatmap function in R [38].

## Ferrozine and Arnow assays

Fe(II) and Fe(III) concentrations were measured with ferrozine assays as described by Jeitner with modifications [39]. Supernatant from lignin-amended and unamended cultures were harvested by removing 15 mL aliquots of culture from serum bottles under anoxic conditions during lag phase, late logarithmic growth phase, and mid-stationary growth phase. Samples were

filtered under anoxic conditions through a 0.45 μm filter to remove biomass and then ultrafiltrated with a 10 kDa Amicon filter. Filtrate (<10 kDa) was tested in triplicate for total iron. Briefly, in a 96-well plate under anoxic conditions, 225 μL of sample with 15 μL 1 M ascorbic acid were added, followed by 60 μL of 50 mg ferrozine mL$^{-1}$ in 500 mM potassium acetate buffer, pH 5.5. Plates were wrapped in tin foil and incubated for 135 min at 37ºC before being measured at 562 nm with a plate spectrophotometer. A separate set of plates had ascorbic acid substituted with water to calculate Fe(II) in the media. For both ferrozine assays a standard curve of Fe(II) was completed as well as samples taken from the abiotic cultures to act as controls.

To quantify catecholates, Arnow assays were completed with the <10 kDa supernatant fraction as previously described [40]. Briefly, 1 mL of a <10 kDa fraction sample was combined with 1 mL 0.5 M HCl, 1 mL nitrite-molybdate reagent, and 1 mL 1 M NaOH. Reactions were incubated for 5 min before being diluted 5-fold with water in a 96-well plate and light absorbance was read at 510 nm. Triplicate technical replicates were taken from each of the three biological replicates as well abiotic controls for each condition in duplicate. A standard curve was produced using 3,4-dihydroxybenzoate [40]. Arnow results of the lignin condition between cultures and abiotic controls across time were analyzed with a two-way ANOVA using the stats package in R [38].

## Inductively coupled plasma (ICP) spectroscopy of kraft alkali lignin substrate

One gram of alkali lignin, low sulfonate (Sigma Aldrich, CAS Number 8068-05-1) was sent for analysis in triplicate to the University of Massachusetts Amherst Soil and Plant Nutrient Testing Laboratory. Lignin was acid wet digested in nitric acid, hydrochloric acid, and hydrogen peroxide in a block digester and measured with ICP Spectroscopy to determine the total P, K, Ca, Mg, Zn, Cu, Mn, Fe, and B.

## Results and discussion

### Physiology and global expression changes in response to lignin

To determine the role of lignin in the anaerobic metabolism and growth of *T. lignolytica* BRL6-1, we compared cultures grown with glucose that were either amended or unamended with lignin. Lignin-amended cultures had a shorter lag phase (5.0 ± 0.6 hrs) compared to cells grown in unamended conditions (11.0 ± 4 hrs, p-value = 0.03). The shortening of the lag phase is expected for diauxic microbes as an adaptive trait for frequently changing environments [41]. Lignin-amended cultures also had higher yields (0.140 ± 0.013 OD$_{600}$) compared to cells grown in unamended conditions (0.124 ± 0.003 OD$_{600}$, p-value = 0.04). Maximum growth rate in unamended conditions was not significantly different from lignin-amended conditions (0.039 ± 0.03 OD$_{600}$ hr$^{-1}$ compared to 0.03 ± 0.008 OD$_{600}$ hr$^{-1}$. Our growth results support previous findings that under lignin-amended conditions, BRL6-1 fitness improves, experiencing a shorter lag phase and achieving a higher final biomass when lignin is amended [25].

To explain this change in growth, Billings *et al.* originally hypothesized that lignin may serve as a secondary C source as well as a potential energy source [25]. To explore this hypothesis, we compared the cellular protein abundance between lignin-amended conditions to unamended. EdgeR analysis uses the negative binomial distribution to detect proteins that are differentially abundant between the two growth conditions [34]. The analysis detected a total of 41 proteins with significantly higher abundance and 101 with lower abundance relative to unamended conditions in late logarithmic phase and a total of 9 proteins with significantly higher abundance and 9 proteins lower abundance at mid-stationary phase (Fig 1).

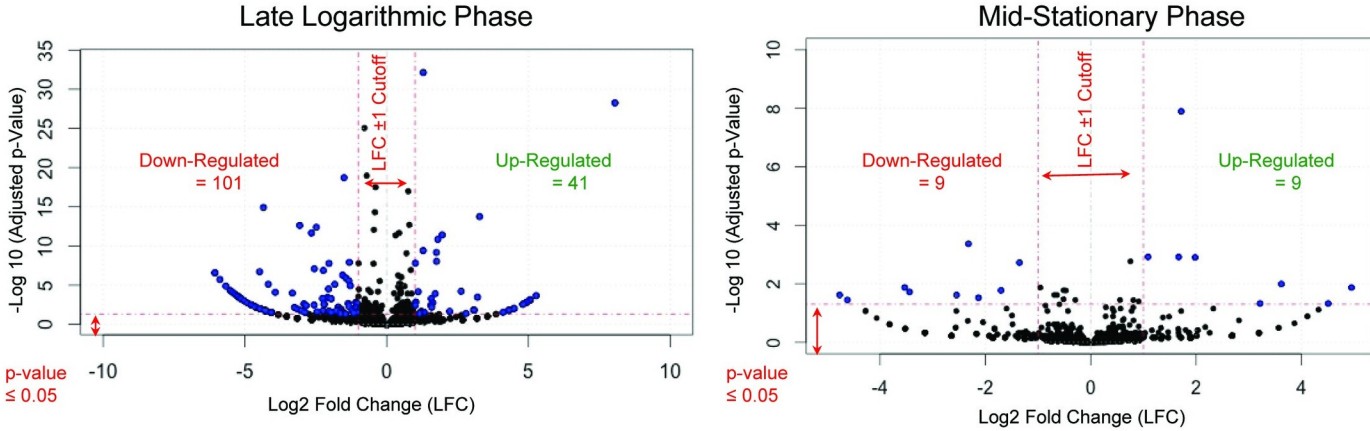

**Fig 1. Cellular protein abundances of *T. lignolytica* BRL6-1 grown in lignin-amended relative to unamended conditions.** Blue dots represent significant change in protein abundance in lignin-amended conditions (p-value < 0.05) relative to unamended conditions whereas black dots denote represent proteins that did not change between lignin-amended and unamended conditions.

### Evidence of proton-relay system-related enzyme for lignin degradation

From our edgeR analysis results, we first investigated the roles of proteins that were significantly higher abundance in lignin-amended cultures relative to unamended in both late logarithmic phase and mid-stationary phase. This subset of proteins included the protein with the highest abundance listed in lignin-amended conditions in both phases, which was annotated as a carboxymuconolactone decarboxylase (CMD) family protein (Pfam 02627). The CMD family of proteins is associated with aromatic degradation in aerobic bacteria, via the protocatechuate branch of the β-ketoadipate pathway, as well as antioxidant defense via peroxidase activity [42, 43]. The CMD family protein identified in BRL6-1 contained an alkylhydroperoxidase (AhpD) domain and CXXC motif ($\log_2$ fold-change of 8 and 7, respectively) (Fig 2A). Based on the CXXC motif, it is thought that the protein detected in BRL6-1 has AhpD-like activity [42]. AhpD is part of an antioxidant defense system that forms a complex with peroxiredoxin, AhpC. The function of AhpD is to restore the enzyme activity of AhpC via reduction [44]. Looking further into the genome, BRL6-1 contains a gene annotated as AhpF, which is an alternative alkyl hydroperoxide reductase to AhpD as seen in *Salmonella typhimurium* (63.5% sequence identity with NCBI BLASTp) [45]. Based on this information, AhpC likely forms a complex with AhpF. Softberry BPROM [46] predicted a promoter site upstream of both AhpC and AhpF in the BRL6-1 genome, suggesting that they are co-expressed and provides further evidence for AhpCF complex formation. Additionally, our edgeR analysis determined AhpCF expression was not significantly different between lignin-amended and unamended conditions. Therefore, this AhpD-like protein's role is likely not to restore AhpC activity but instead could reduce other substrates near the membrane surface when lignin is present.

AhpD uses is a proton relay system to reduce its substrates [47], which was originally described as protons being shuttled from the active site of a protein to bulk solvent molecules [48]. This mechanism has been observed previously in the lignin degrading enzyme, LigL, found in *Sphingomonas paucimobilis* SYK-6. LigL catalyzes stereospecific oxidation of the benzylic alcohol as the first degradation step of lignin-derivative, (αS, βR)-GGE [49, 50]. The proton relay mechanism has also been described for p-Cresol methylhydroxylase (PCMH) in *Pseudomonas* species to degrade phenol p-cresol as well as p-hydroxybenzyl alcohol [47]. Therefore, this AhpD-like protein could be reducing lignin-derived compounds in the cell

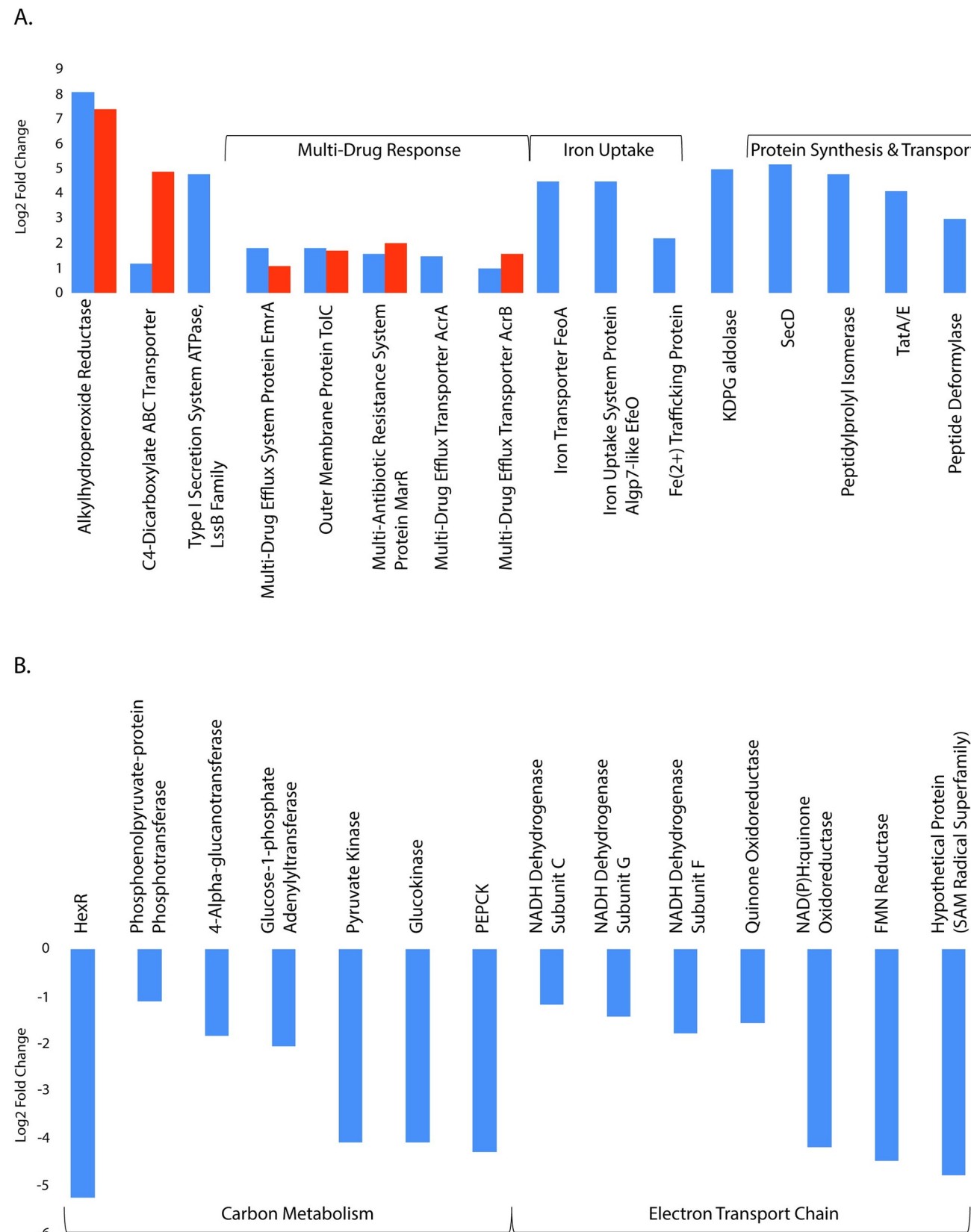

**Fig 2. Cellular proteins with significant change in abundance (log$_2$ fold change, p-value < 0.05) related to carbon metabolism and energy production by *T. lignolytica* BRL6-1 under lignin-amended conditions relative to unamended.** (A) Cellular proteins of BRL6-1 with significantly high and (B) low abundance under lignin-amended conditions compared to lignin unamended (p-value < 0.05). Abbreviations are the following: phosphoenolpyruvate carboxykinase (PEPCK); 2-keto-3-deoxy-6-phosphogluconate (KDPG) aldolase.

using this mechanism. In support of cellular aromatic compound uptake, a C4-dicarboxylate ABC transporter protein was higher in abundance in lignin-amended conditions in both late logarithmic and mid-stationary phase (Fig 2A; log$_2$ fold-change of 1.2 and 4.9, respectively). This transporter has an 87% sequence identity to transporter, DctA, in *Pseudomonas chlorora-phis* O6 that was found to be essential for benzoate uptake [51]. In addition, indicator species analysis of the metabolome had benzoic acid significantly higher in abundance within BRL6-1 cells at late logarithmic phase compared to mid-stationary phase (Fig 3). Further study should

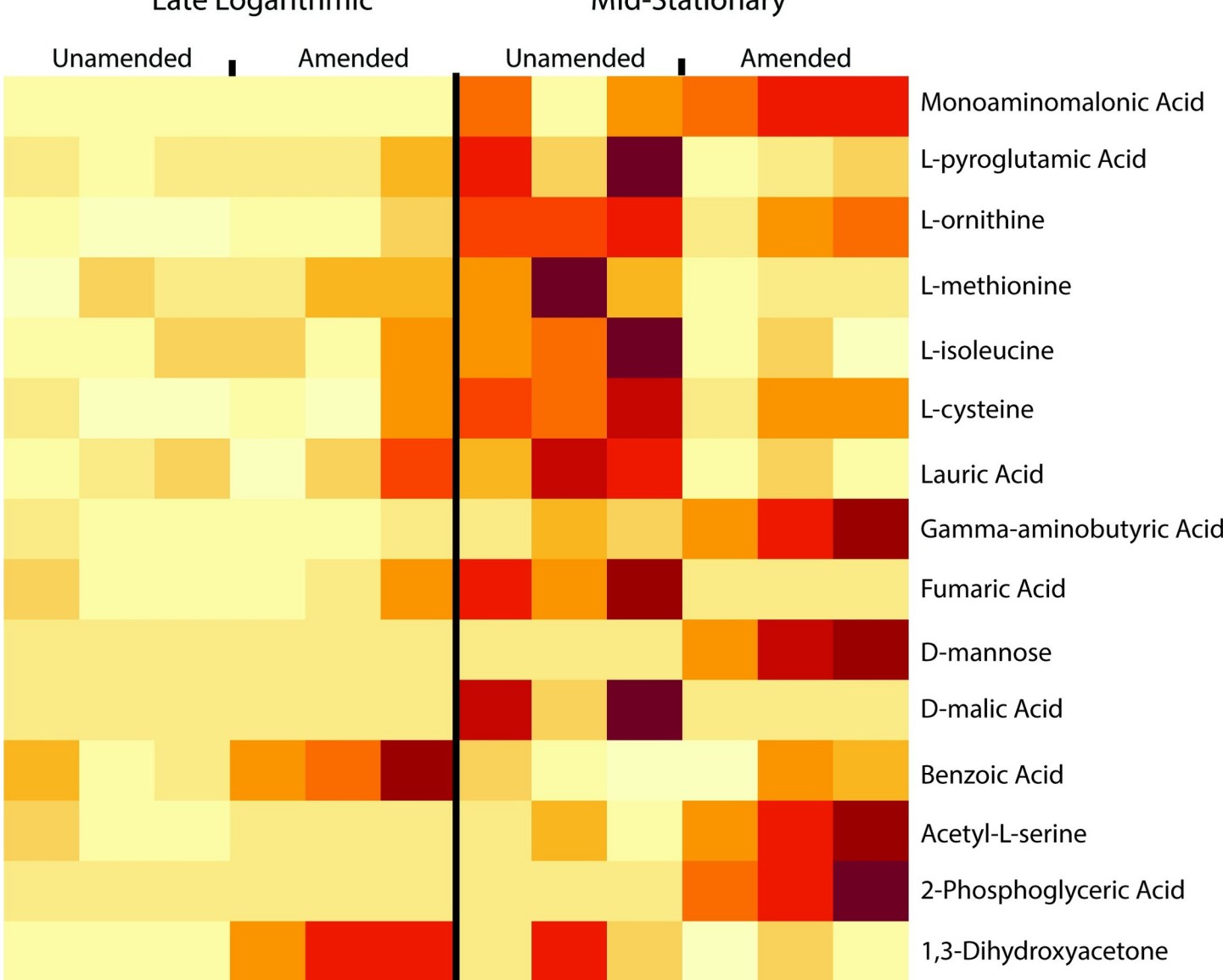

**Fig 3. Heat map of indicator metabolites in lignin-amended and unamended cultures for both late logarithmic and mid-stationary phase.**

focus on elucidating the molecular mechanisms to define whether AphD has a direct role in lignin degradation.

## Lignin-amended cultures shift in carbon metabolism and energy production

Our differential protein abundance data as well as metabolomic indicator species analysis suggest the potential transport of benzic acid or benzoate-like compounds into the cell and their reduction via AphD. Secondary carbon utilization could explain the BRL6-1 growth advantage in the presence of lignin. If lignin is being depolymerized and transported into the cell as a secondary carbon source, we expect to see an increased protein abundance in enzymes related to central carbon metabolism. For example, *Sphingomonas paucimobilis* SYK-6 relies on the tricarboxylic acid cycle (TCA) and gluconeogenesis pathways in the presence of vanillin for energy metabolism [52]. We also expect that proteins homologs of the ferulic acid metabolic pathway in BRL6-1 would be higher in abundance in the presence of lignin [25]. The ferulic acid metabolic pathway transforms ferulic acid, a lignin derived monomer, to β-ketoadipate, with vanillate and protocatechuate as intermediate compounds. Therefore, to explore whether there was further evidence for secondary carbon utilization, we focused on any significantly differentially abundant proteins relating to carbon (C) metabolism in the presence of lignin. In contrast to our expectations, we found that glycolysis and gluconeogenesis to be repressed and no differential expression of the ferulic acid pathway was detected in the presence of lignin. In addition, while there was a significant growth phase effect (p = 0.007) as well as interaction effect (treatment and growth phase; p = 0.016) on metabolite abundance (Fig 4), the indicator species analysis did not identify any ferulic acid metabolic pathway intermediates as significantly higher in either late logarithmic or mid-stationary phase in the presence of lignin (Fig 3).

Looking at differential gene expression in the central carbon metabolism in the presence of lignin, we found evidence for a switch from glycolysis in unamended growth to the Entner-Doudoroff (ED) pathway in lignin-amended growth. BRL6-1's glucokinase and pyruvate kinase, which are responsible for the first and last step of glycolysis, respectively, were lower in abundance by a $\log_2$ fold-change of -4, suggesting that the lignin-amended conditions lead to the repression of glycolysis by late logarithmic phase (Fig 2). Indicator species analysis revealed that related glycolysis compounds, 1,3-dihydroxyacetate, D-mannose, and 2-phosphoglyceric acid, were significantly higher in abundance in lignin-amended conditions. These pools of intermediates suggest they are a result of glycolysis slowing down. For example, with glucokinase being lower in abundance in lignin-amended conditions, mannose cannot be phosphorylated and would accumulate in the cell [53, 54]. Phosphoenolpyruvate carboxykinase (PEPCK), the rate limiting enzyme for gluconeogenesis, was also log2 fold-change of 4 lower in abundance in lignin-amended growth conditions compared to unamended growth conditions. This suggests that PEP was not being funneled into gluconeogenesis in the presence of lignin. We did, however, observe a 2-dehydro-3-deoxy-phosphogluconate (KDPG) aldolase that was significantly higher in abundance by a $\log_2$ fold-change of 5 in lignin-amended growth conditions compared to unamended conditions (Fig 2). These findings suggest that in the presence of lignin, there is a higher conversion of KDPG to pyruvate via the Entner–Doudoroff (ED) pathway in BRL6-1. This shift from glycolysis to the ED pathway by BRL6-1 is a different response compared to other bacteria capable of catabolizing lignin or lignin derived compounds. This includes *Sphingomonas paucimobilis* SYK-6 which lacks enzymes for a complete ED pathway [52], *Enterobacter lignolyticus* SCF1, whose glycolysis related genes were upregulated in the presence of lignin [23], and *Pseudomonas putida* KT2440, which relies on the β-ketoadipate pathway in the presence of benzoate [55].

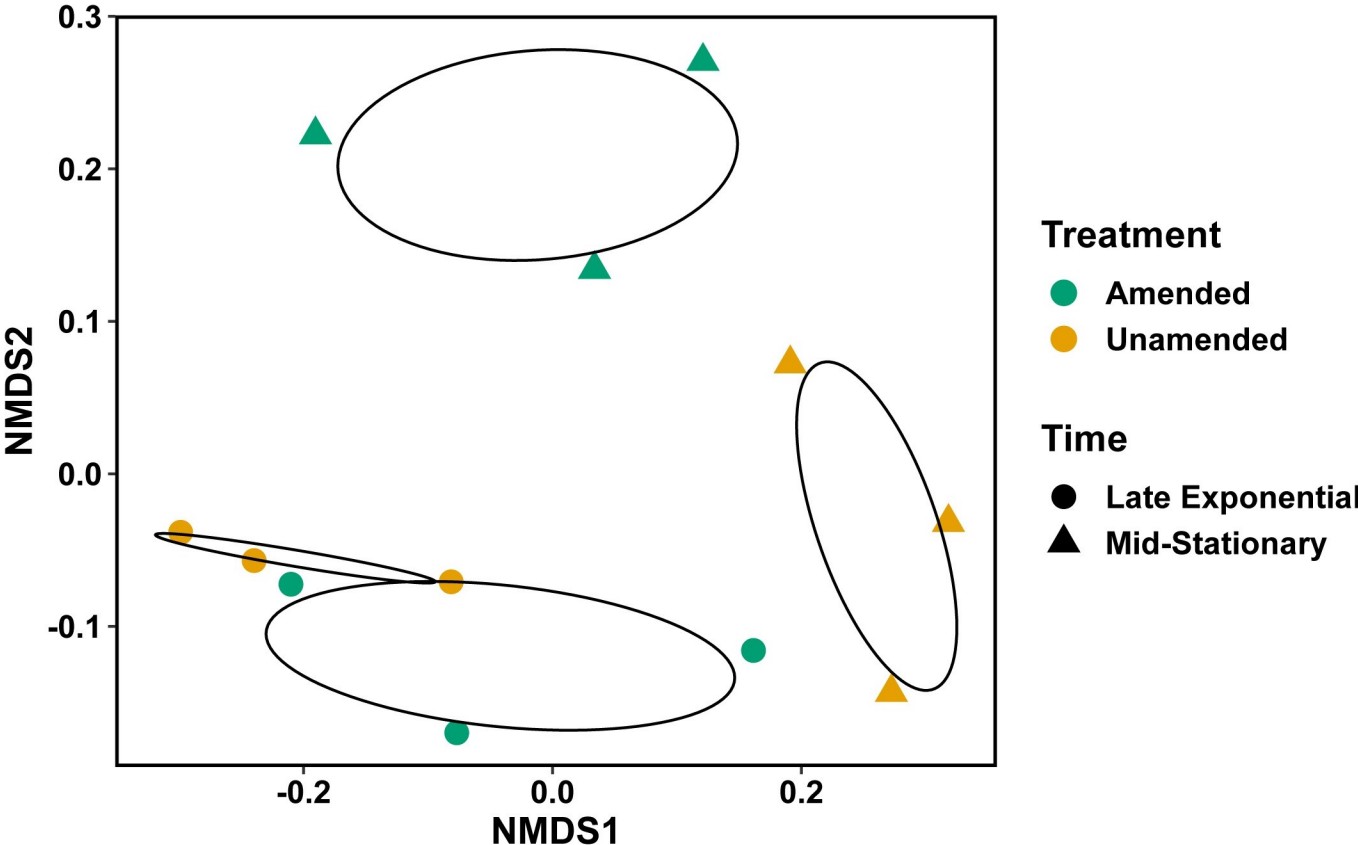

**Fig 4. Non-metric multidimensional scaling (NMDS) ordination of metabolites in lignin-amended and unamended cultures for both late logarithmic and mid-stationary phase.** Green represents lignin-amended and yellow represents lignin unamended. Circle indicates late logarithmic phase and triangle represents mid-stationary phase.

In further support of alternative glucose metabolism in the presence of lignin, we detected possible regulatory changes in carbohydrate metabolism in lignin-amended growth. HexR is a global transcription factor utilized for fine-tuning the carbohydrate catabolic pathways to adapt to variable C availability in the environment [56–58]. In lignin-amended conditions, the HexR transcriptional factor was significantly lower in BRL6-1 by a $\log_2$ fold-change of -5 during late logarithmic growth phase (Fig 2B). HexR is a global transcription factor utilized for fine-tuning the carbohydrate catabolic pathways to adapt to variable C availability in the environment [56–58]. It has been previously shown that HexR is present in several groups of Proteobacteria and regulates a rather complex set of operons [58]. Leyn and colleagues (2011) deleted *hexR*, which resulted in the de-repression of transcription of the central glycolytic genes as well as the activation of genes involved in gluconeogenesis.

The indicator species analysis identified many TCA intermediates significantly higher in abundance in BRL6-1 when grown in unamended conditions during mid-stationary phase in contrast to lignin-amended conditions (Fig 3). This suggests that the TCA intermediates are being turned over faster in lignin-amended conditions. Further study is required to determine how the presence of lignin is increasing the flux of the TCA cycle in recycling intermediates.

To determine if lignin had a role in energy production, we searched for enzymes that were related to the electron transport chain that had differential abundance in lignin-amended conditions compared to unamended. During late logarithmic phase, 6% of all significantly lower

abundance proteins ($\log_2$ fold-change $>1$, p-value $<0.05$) in the presence of lignin were NADH dehydrogenase subunits ($\log_2$ fold-change between -1.2 to -4) and one flavin mononucleotide ($\log_2$ fold-change of -4) (Fig 2B). This suggests that in the presence of lignin, BRL6-1 is relying on an alternative means to obtain energy, and so we investigated whether iron reduction could be a supplemental energy source in lignin-amended growth.

## Iron redox by *T. lignolytica* BRL6-1 in the presence of lignin and hypothesized mechanisms

A substantial portion (14%) of the proteins that were significantly higher in abundance ($\log_2$ fold-change $>2$) in the presence of lignin during late logarithmic phase related to Fe(II) reduction (Fig 2A). We hypothesized that BRL6-1 could be obtaining energy using iron reduction and that lignin is increasing the access to iron in the media due to its strong affinity for iron [59]. Lignin binds iron and makes it more soluble in the environment but not necessarily more bio-available for cellular use [60]. This considered, BRL6-1 may have a mechanism that is disrupting the lignin-iron association. By doing so, BRL6-1 could obtain both iron and a potential C source faster than cells in unamended conditions, explaining the ability of BRL6-1 to exit lag phase more quickly in the presence of lignin [61].

To further investigate the relationship between lignin, iron, and BRL6-1 fitness, we first asked if lignin was an additional iron source aside from the SL-10 minerals we added to the growth media. An increase in bioavailable iron to the cells should result in higher fitness if iron was limiting [62]. ICP spectrophotometry of the lignin substrate measured 38 ppb Fe that would not be present in the unamended conditions. Therefore, to test whether the 38 ppb Fe was benefitting BRL6-1 fitness between lignin-amended and unamended conditions, we conducted Fe-addition growth curve experiments and monitored BRL6-1 growth. Compared to cells grown in lignin unamended conditions reported above, lignin unamended conditions with a 38 ppb iron addition were not significantly different in lag phase, growth rate, or biomass yield: 10.5 ±5 hrs, 0.072 ±0.03 $OD_{600}$ per hr, 0.124 ±0.007 $OD_{600}$, respectively. These results suggest that additional iron alone was not enough to benefit BRL6-1 growth.

To test if lignin has a strong affinity for iron in our system, we completed a ferrozine assay for $<10$ kDa fractions of supernatant from lag phase, late logarithmic growth phase, and mid-stationary growth phase. We expected that if the iron was binding to lignin in the media, we should see less bioavailable iron in the supernatant of lignin-amended compared to lignin unamended conditions. In lignin-amended conditions, Fe(III) was not detectable in the $<10$ kDa fractions throughout the entire growth curve whereas 292 ppb Fe(III) was detected during lag phase in lignin unamended conditions. As bacterial biomass increased over time, Fe(II) accumulated in both conditions to similar concentrations (Fig 5). There was no change in Fe(II) and Fe(III) concentrations in abiotic controls. This suggests that Fe(III) was bound to the lignin and was reduced to Fe(II) by BRL6-1.

With evidence for Fe(III) bound to lignin and BRL6-1 Fe(III) reduction to Fe(II) based on ferrozine assays, we next wanted to determine the mechanism that BRL6-1 may use to acquire and reduce iron. Siderophores are organic molecules used by bacteria to chelate and acquire Fe(III) under iron-limiting conditions [63]. The most common siderophore used by bacteria are catecholates, which rely on hydroxyl groups of the catechol rings to form the iron chelation center [40]. To see if BRL6-1 produces this type of siderophore, we completed Arnow assays on $<10$ kDa supernatant fractions from lignin-amended and unamended cultures during lag phase, late logarithmic growth phase, and mid-stationary growth phase. Catecholate detection was seen only in lignin-amended conditions; however, there was no change in concentration of catechol over the course of the growth curve (Fig 6). Additionally, abiotic controls of the

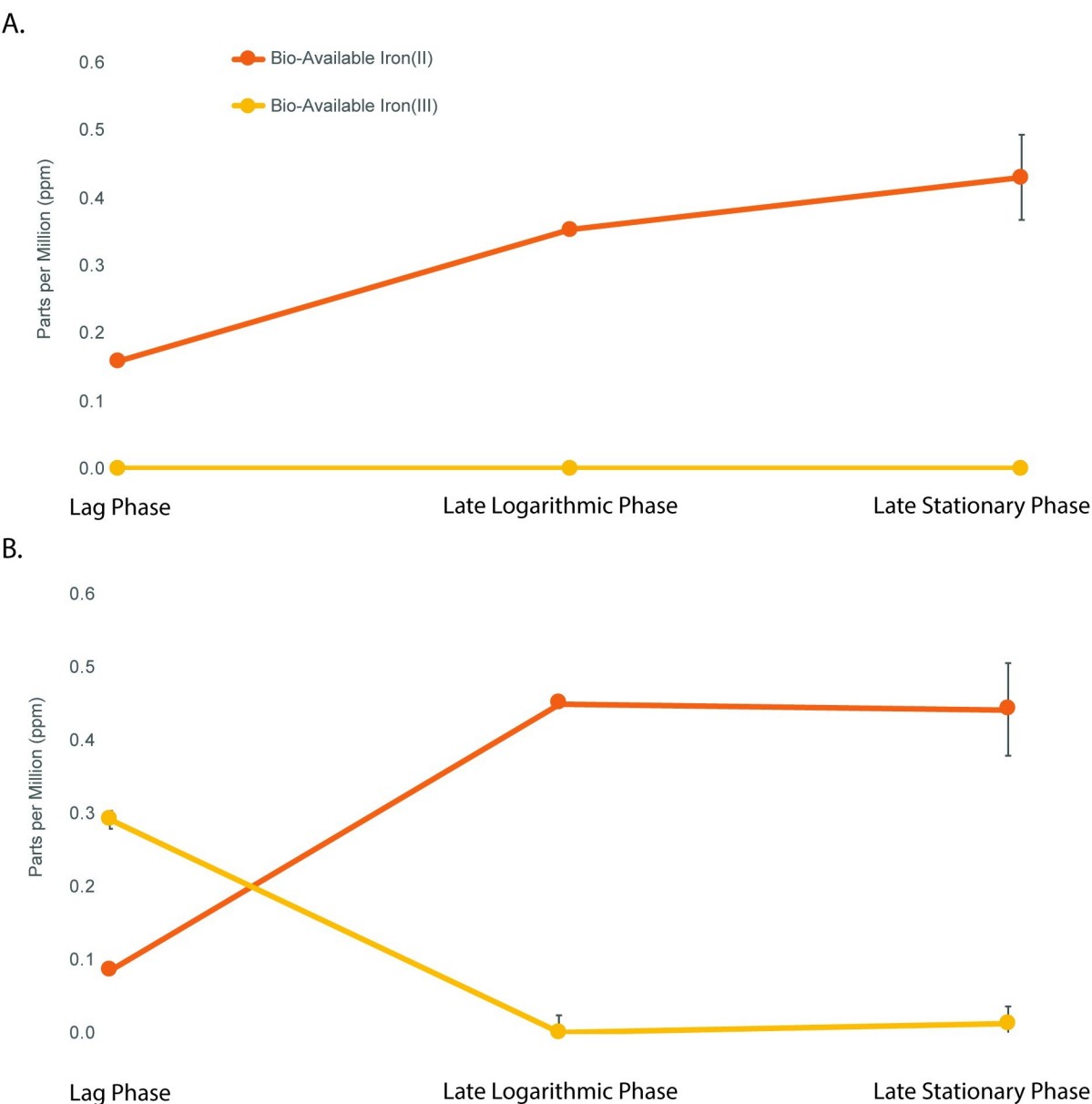

**Fig 5. Bio-available Fe(II) (orange) and Fe(III) (yellow) concentrations in parts per million (ppm).** Mean concentrations of Fe(II/III) (± SD) when BRL6-1 was cultured in (A) lignin-amended and (B) in unamended conditions at lag phase, late exponential phase, and late stationary phase (n = 3 for each treatment).

lignin-amended conditions had similar concentrations to biotic replicates. Two-way ANOVA analysis supported this finding with no significant effect between the cultures and abiotic controls (p = 0.159), growth phases (p = 0.851), or their interaction (p = 0.308). This is likely due to Arnow assays being non-specific between catecholates and compounds containing catechol, such as soluble lignin [64], making it difficult to differentiate sources as well as any small changes in concentration of such metabolites. BRL6-1 may also be producing other groups of siderophores such as hydroxamates or carboxylates [63], which would need to be detected with a Csáky assay or the use of phenolphthalein and sodium hydroxide, respectively [65, 66]. Despite these limitations, initial results suggest that siderophores are unlikely to be the main

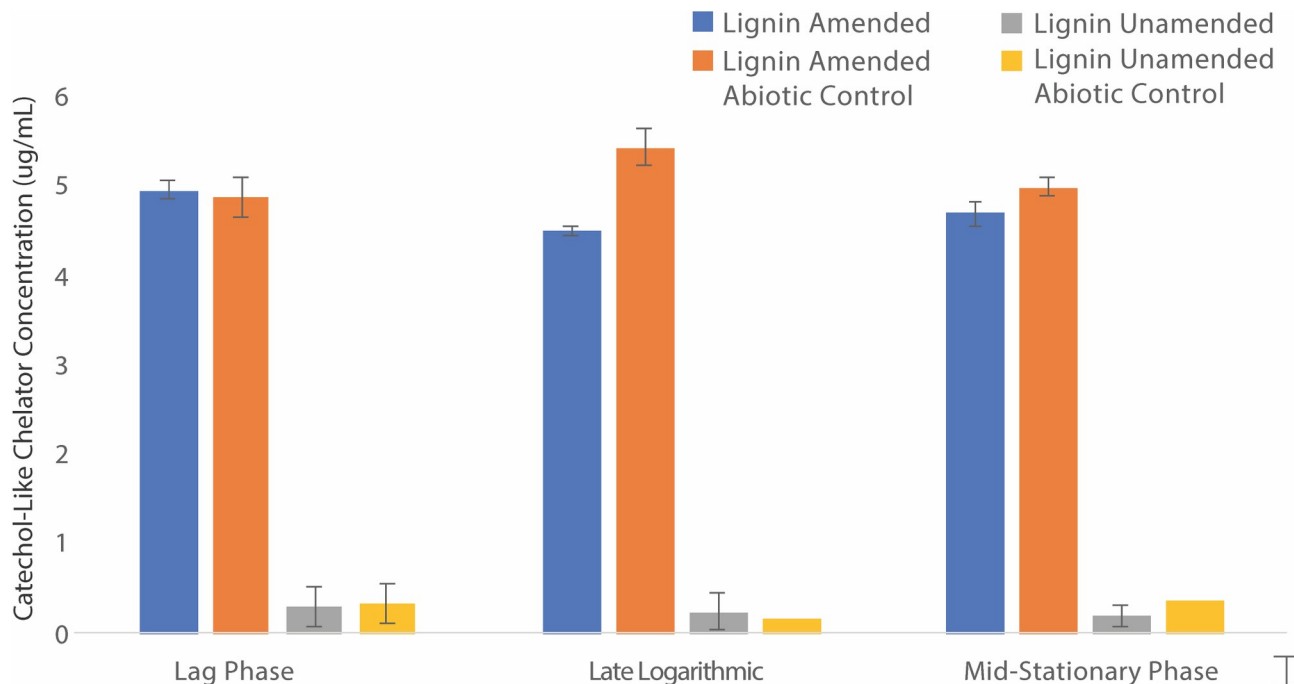

**Fig 6. Arnow assay of catechol-like compounds present in supernatant.** Catechol-like chelator concentrations ($\mu$g/mL; ± SD) at lag, late logarithmic, and mid-stationary phase of BRL6-1 growth under lignin-amended (blue) and unamended conditions (gray). Abiotic controls are striped for both conditions (n = 3 for each treatment, n = abiotic controls).

explanation for the Fe redox being detected by the ferrozine assay and instead another mechanism may be used by BRL6-1.

We investigated to see if BRL6-1 secreted lignolytic enzymes or iron reducing proteins in the presence of lignin as seen for aerobic fungi and bacteria [12, 13, 15]. Samples from late stationary phase were run on an SDS-PAGE and silver stained to detect differential banding between the two growth conditions. Differential banding was observed at 20kDa under lignin-amended conditions (Fig 7). In addition, because BRL6-1 has had previously predicted peroxidases in its genome [25] and that lignin peroxidases are 35–48 kDa [67], we also were interested in the bands at 37 and 50kDa. Therefore, bands were cut out at 50 kDa, 37 kDa, and 20 kDa for both conditions to identify the proteins present by sequencing. Banding was also seen in lignin-amended conditions at approximately 11kDa and 8kDa. However, these are likely artifacts from the lignin phenolics interacting with the proteins that were not removed during the TCA precipitation [68, 69], and therefore cannot be further investigated with confidence.

A protein originally annotated as hypothetical, WP_024871222.1 was detected in all three lignin-amended biological replicates with a predicted size of 20.8 kDa. There were no conserved domains detected in this protein, but Position-Specific Iterated (PSI) BLAST analysis of the protein identified homology to a hypothetical protein from *Alteromonadales bacterium* BS08 (53% Identity; E-value 4e-61). BS08 was isolated from the interlamellar junction of a *Bankia setacea*'s gill. *B. sectacea* are also known as shipworms, that digest wood as a food source [70]. These organisms rely on their gill endosymbionts, such as BS08, for cellulytic and proteolytic enzymes in order to digest the wood. It is possible that the hypothetical protein in BS08 as well as in BRL601 also are active in lignocellulosic break down. Additionally, WP_024871222.1 had homology to enzymes in the radical S-adenosyl-L-methionine (SAM) superfamily (37.7% Identity; E-value 0.06), a wide range of enzymes that support oxygen-

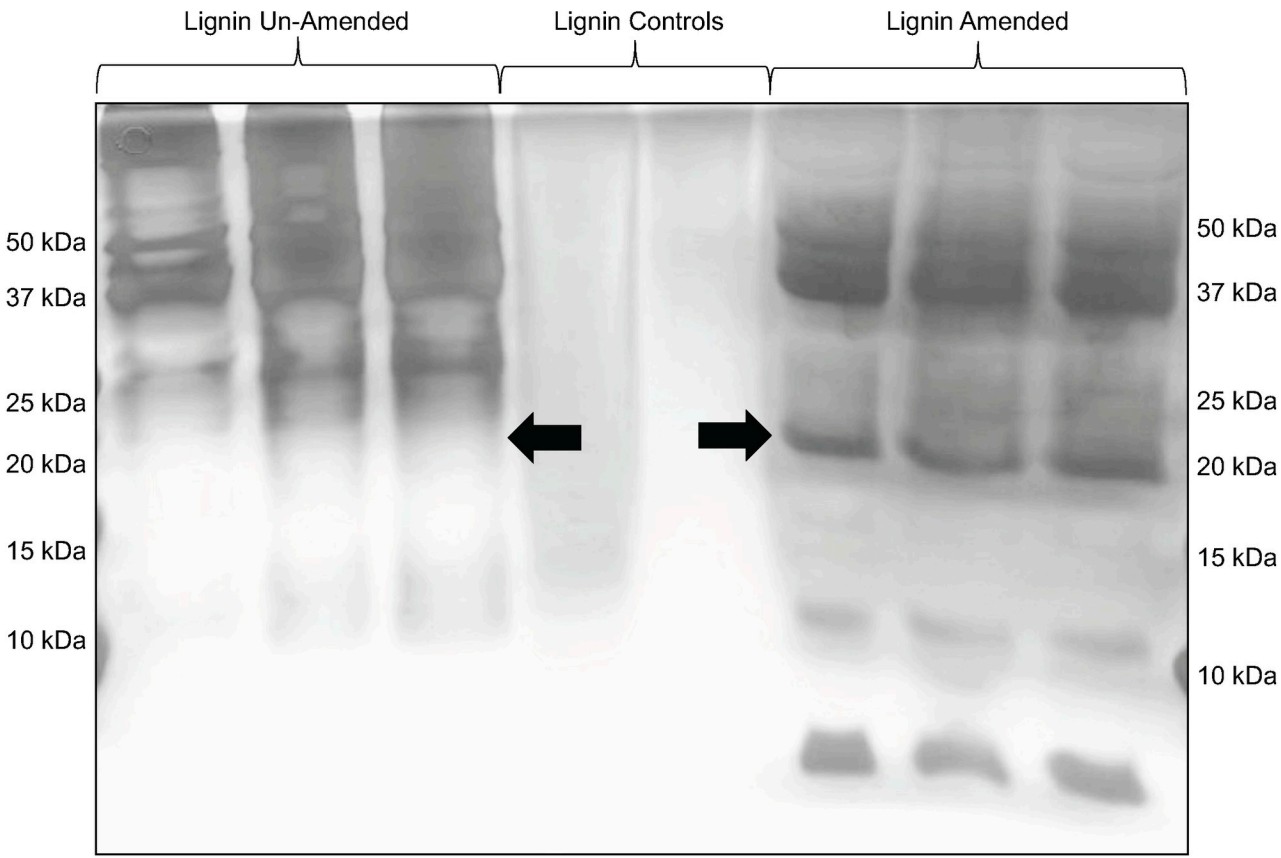

**Fig 7. SDS-PAGE of secretome of *T. lignolytica* BRL6-1 cultured in lignin-amended and unamended conditions.** Arrows showing differential banding at 20kDa.

independent alternatives to aerobic pathways [71]. This may be relevant since *Tolumonas lignolytica* BRL6-1 as other *Tolumonas* spp., are the only members of the *Aeromonadaceae* that are oxidase and catalase negative [25]. They may rely on alternative means, such as a radical SAM protein, to circumvent radical stress produced by the additional effort that the metabolism has to do in order to oxidize a complex structured high redox potential substrate, such as lignin, or by coping during fluctuating redox potentials in soil. Mediators or co-oxidants not only increase the catalytic ability of these enzymes, but also largely expand their substrate scope to those with higher redox potential or more complicated structures. This is especially true for BRL6-1, which was isolated from tropical soils known to have frequent strong fluctuations in redox [72, 73]. Therefore, it is possible, that WP_024871222.1 has a role in lignin modification via radical formation, but more research would be needed to link this putative SAM homolog to anoxic lignin transformation.

## Conclusions

The presence of lignin has been previously shown to be beneficial for the growth of *T. lignolytica* BRL6-1 [25] with speculation that lignin was acting as a secondary C source or energy source. In our study, cellular proteomic analysis revealed that in the presence of lignin there is a shift from glycolysis to the Entner-Doudoroff pathway in late logarithmic phase. In addition, our metabolomic data suggested that benzoic acid is present in BRL6-1 during late logarithmic phase and that there is a significant treatment effect on the metabolite abundances during

mid-stationary phase. During mid-stationary phase, unamended cultures have significantly higher TCA intermediates present compared cultures grown in the presence of lignin. This suggests that in the presence of lignin, tricarboxylic acid cycle intermedites have a higher turnover in the cells. Further study, such as a $^{13}$C metabolic flux analysis, is recommended to aid in resolving how lignin could be acting as a secondary carbon source and through which pathways.

The cellular proteomic analysis also detected that 14% of the upregulated proteins by $\log_2$ fold-change of 2 or greater related to Fe(II) transport in lignin-amended cultures compared to unamended. Transient iron accumulation in *Salmonella enterica* serovar Typhimurium is required in order for the cells to come out of lag phase [61] and so it was possible that the up-regulation of iron enzymes might be due to BRL6-1 having the same iron requirement or relied on iron redox for energy [74]. However, lignin has a strong affinity for iron [59] and therefore we would expect that iron is less bioavailable to the cells in lignin-amended conditions [60]. Ferrozine assays of the <10kDa supernatant fractions confirmed that Fe(III) was bound to lignin, but it was reduced to Fe(II) when BRL6-1 was present, suggesting redox activity by the cells. To explain this redox activity, we hypothesized that BRL6-1 is producing a protein that acts as both an iron chelator and redox agent under anoxic conditions to obtain the iron bound to lignin. Secretome (extracellular enzyme) analysis coupled with LC-MS/MS identified the presence of a protein of unknown function but had homology to enzymes in the radical SAM superfamily, suggesting that it may have a role in radical formation in lignin-amended conditions.

While our work illustrates a potential molecular mechanism for anaerobic lignin modification by *T. lignolytica* BRL6-1, protein isolation and characterization are needed to confirm that this protein interacts with the Fe(III) bound to lignin and reduces it to Fe(II) for cellular use. Further analysis with electron paramagenetic resonance (EPR) as well as nuclear magnetic resonance (NMR) are necessary in order to confirm that organic radicals are being produced in the process of this enzyme obtaining iron and that these radicals alter the structure of lignin. By continuing research into the biochemical nature of microbial lignin transformations under anaerobic conditions, industries using lignocellulose as raw material will be one step closer to lignin valorization.

## Supporting information

**S1 Raw images. Undoctored image of silver stained gel.** Silver stained SDS-PAGE of T. lignolytica BRL6-1 secretome cultured in unamended conditions (lanes 2–4) and lignin-amended (lanes 7–9). Lanes 1 and 10 are Precision Plus Protein Dual Color Standards (250-10kDa). Lanes 4 and 5 are abiotic controls of lignin-amended conditions. Image was taken with flat bed scanner. Figure in manuscript was submitted as gray scale with ladder lanes cropped (X) and substituted for labels instead. Figure is labeled as Fig 7.
(PDF)

## Acknowledgments

We are grateful to Steven Eyles for assistance with mass spectral data at the University of Massachusetts Mass Spectrometry Center. This publication was also developed with assistance from the U.S. Environmental Protection Agency, but it has not been formally reviewed by EPA. The views expressed in this document are solely those of the authors and do not necessarily reflect those of the Agency. EPA does not endorse any products or commercial services mentioned in this publication.

## Author Contributions

**Conceptualization:** Gina Chaput, Roberto Orellana, Blake Simmons, Kristen M. DeAngelis.

**Data curation:** Gina Chaput, Joshua N. Adkins, Carrie D. Nicora, Young-Mo Kim, Rosalie Chu.

**Formal analysis:** Gina Chaput, Carrie D. Nicora, Young-Mo Kim.

**Funding acquisition:** Gina Chaput, Blake Simmons, Kristen M. DeAngelis.

**Investigation:** Gina Chaput, Andrew F. Billings, Lani DeDiego, Joshua N. Adkins, Carrie D. Nicora, Young-Mo Kim, Rosalie Chu.

**Methodology:** Gina Chaput, Carrie D. Nicora, Kristen M. DeAngelis.

**Project administration:** Kristen M. DeAngelis.

**Resources:** Kristen M. DeAngelis.

**Supervision:** Kristen M. DeAngelis.

**Validation:** Kristen M. DeAngelis.

**Visualization:** Gina Chaput.

**Writing – original draft:** Gina Chaput, Kristen M. DeAngelis.

**Writing – review & editing:** Gina Chaput, Andrew F. Billings, Roberto Orellana, Joshua N. Adkins, Kristen M. DeAngelis.

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
