## [Decision Letter · Decision Letter 0]

8 Jun 2020

PONE-D-20-14077

Iron Chelator-Mediated Anoxic Biotransformation of Lignin by Novel sp., Tolumonas lignolytica BRL6-1

PLOS ONE

Dear Dr. DeAngelis,

Thank you for submitting your manuscript to PLOS ONE. After careful consideration, we feel that it has merit but does not fully meet PLOS ONE’s publication criteria as it currently stands. Therefore, we invite you to submit a revised version of the manuscript that addresses the points raised during the review process.

The reviewers offered thoughtful and thorough comments, many of which centered on the clarity and completeness of the the experimental conditions. I concur with the reviewers on these issues and ask that your revisions address their comments, point-by-point. I also agree with reviewer #2's concerns about overstating the findings in the conclusions and the title

Beyond the reviewers specific criticisms, I was puzzled by the choice of lignin used throughout the study. In the absence of additional analyses, can you be certain that Sigma's kraft lignin is free of contaminants, e.g. hemicelluloses?  I noticed some elemental analysis (lines 165-169), but no other compositional data. Is this available from Sigma? Obviously, synthetic lignins would be preferable, but this would seem a major undertaking. If appropriate data is unavailable, minimally the text should clarify the design's limitations.

We look forward to receiving your revised manuscript.

Kind regards,

Daniel Cullen

Academic Editor

PLOS ONE

Journal Requirements:

4. Please amend the manuscript submission data (via Edit Submission) to include author Carrie Nicora.

5. Please include captions for your Supporting Information files at the end of your manuscript, and update any in-text citations to match accordingly. Please see our Supporting Information guidelines for more information: http://journals.plos.org/plosone/s/supporting-information

Reviewers' comments:

Reviewer's Responses to Questions

**Comments to the Author**

1. Is the manuscript technically sound, and do the data support the conclusions?

Reviewer #1: Partly

Reviewer #2: Yes

2. Has the statistical analysis been performed appropriately and rigorously? 

Reviewer #1: Yes

Reviewer #2: No

3. Have the authors made all data underlying the findings in their manuscript fully available?

Reviewer #1: Yes

Reviewer #2: No

4. Is the manuscript presented in an intelligible fashion and written in standard English?

Reviewer #1: Yes

Reviewer #2: No

5. Review Comments to the Author

Reviewer #1: Manuscript appears to be of significant quality and relatively well written. Using “omics” and complementary biochemical studies the manuscript paints a wonderful picture of what appears to be a novel bacterial anoxic lignin utilization system. Although nice, everything is circumstantially connected and still requires the difficult task of detailed biochemical analysis of the proteins involved. As such, the title comes off as overselling what has actually been done. I suggest retitling.

Also, I think that the methods need more detail.

Also, while the presentation is nice, I think the results shown are limited. When I look at the SDS-PAGE I ask what of the 11kDa and 8kDa bands? If just artifacts there should be a comment. These were >10kDa fraction correct? So what are these bands?

Almost all figures would benefit by more contrasting colors selection blue/light blue or black/blue. What about orange/blue or red/green? This included figure 4. The patterns should be replaced with colors.

Abstract:

Line 27: “10kDa fraction” is not important information for the abstract.

Introduction:

Last paragraph: Does not really serve the purpose of the typical closing paragraph of an “Introduction”. ‘In this work we…..’ is missing. Instead a hypothesis is introduced, which seems to be the working hypothesis in consideration of the proposed research results.

Materials and Methods:

General note: Bits of the Materials and Methods section (often the first sentence of a paragraph) out of place and read more like the Results section.

Line 109: change “for” to “of”.

Proteomics section, Lines 109-133: The section appears to be explain everything twice. Once for the cell pellet and then again for the spent growth media. Perhaps this can be consolidated. I believe data processing was likely done the same in both cases. Consider merging the two paragraphs into one.

Line 112: Remove “Briefly described”.

Line 113: Could more information or a reference be included regarding the method for whole cell protein preparation?

Line 114: “…searched using MS-GF+….”

Line 115: spelling of “Porcine”.

Line 125-126: description of 10kDa filtration is not accurately described. I believe this was a molecular or ultrafiltration centrifugal filter device similar to that described on Line 148. Amicon?

Line 143: Delete “In order to confirm that iron was bound to lignin,”

Line 149-151: All volumes provided for composition, except for the 500 mM buffer.

Line 156: Delete “To determine if BRL6-1 is producing catechol-like siderophores known as catecholates” and replace with something smaller such as “To quantify catecholates …..”.

Line 157: “….as previously described.”

Results and Discussion

Line 175: The “…shorter lag phase….” I think needs more discussion. This is a theme throughout manuscript and I’m not sure that it has been hashed out enough. It seems clear lignin is doing something. It does not seem to be a direct Fe effect based on the results. Instead I believe the authors are pointing to a hypothetical protein of about 20kDa which is actually not directly connected to the other results. Circumstantially perhaps. So then it must the carbon. Then that would mean that the bug is more efficient in (or regulated for preference) in the use of lignin degradation products as compared to glucose.

Line 207: “For our protein….” Should be stated better. Maybe, “This CBM protein….”

Line 227: Figure Legend, KDPG is missing the enzyme descriptor “aldolase”.

Line 237-239: Not complete thought.

Line 260: Line 257 has PEPCK spelled out upon first use in text, but KDPG is not spelled out and I believe it is the first use as well.

Line 262: “dynamic” seems like a strange word. Should it be maybe “unusual”?

Line 269-271: So either way justifies the same outcome?????? That does not sound scientific!

Line 292-293: This sentence and surrounding are not communicated well. Please make more clear.

Line 312: Fe(II/III) should just be written out.

Line 343: “Differential banding was observed at 20kDa….”

Line 356: “…homology to a hypothetical…”

Line 383: Yes, the manuscript definitely illustrates potential of a molecular mechanism. There appears to be a leap at the end to connect all the dots in story. Do not feel as though it is all substantiated. Perhaps a new title and trimmed message may help bridle the wild story!

Line 511: Issue with reference.

Reviewer #2: Where the response given to editorial questions is “no”, this is because the system does not permit a more nuanced response such as “probably with further information”, “needs clarification”, “not yet” or “not entirely”. These would have been my choices otherwise.

Main comments

This is an interesting study of bacterial lignin degradation capabilities. Whereas fungal mechanisms of lignin degradation have become quite well defined, bacterial processes particularly in anaerobic conditions are much less well understood. This study provides some welcome new insights into bacterial mechanisms, with hypotheses suggested for further exploration. Some broader ecological context in the Introduction might be useful too, as the final mineralisation of lignocellulosic substrates typically occurs in soils or sediments.

As these mechanisms are of interest to researchers such as ecologists interested in carbon fluxes and food webs, who are less familiar with microbiological methodology, it would be helpful if the text were made more accessible to non-specialists. Consistently specifying where data derived from pellet or supernatant are being presented would be helpful. Some of the detailed comments below give examples where adjustments could assist in this respect. Also, a sentence or two in the conclusions about the ecological implications of the findings would be useful.

Using biological replicates of proteomic data is a particularly valuable feature of this study. To properly evaluate the statistical treatment of the findings, clarification of details of experimental design in the methods, with linking wording in the Results and figure legends is needed. Some additional analysis may also be worthwhile. A flow diagram of the experimental design showing levels and type (technical and biological) of replication, type of sample (supernatant, pellet) and experiments undertaken would assist the reader. The text would need to be adjusted to link to wording used in this diagram.

Detailed comments and suggestions of edits

Numbers refer to the line numbers in the left margin

24: develops a higher biomass?

66: Mass production……is not possible

70-71: Restructure sentence for increased clarity.

83: worth mentioning that this bacterium was isolated from forest soil (and possibly an anoxic layer within the soil) this point could be briefly addressed in the Conclusions too.

84: …… was demonstrated to have

87: perhaps “BRL6-1 secretes a protein……”?

99: Perhaps “Triplicate cultures were grown anaerobically at 30°C and triplicate uninoculated….. “ Wording needs to be more specific about replication, 3 cultures per experimental condition presumably.

124: 100mL samples from the cultures?

136: Suggest giving a bit more detail about what “spectral counts” means.

144: as described by Jeitner

156: To determine whether BRL6-1 produces…..

157: as described by…… or as previously described

161: absorbance was read

161-162: “Samples……..in triplicate.” Does this mean that triplicate technical replicates were taken from each of three cultures and three abiotic controls, or one biological replicate was taken from each replicate culture and each abiotic control?

175: a shorter lag phase or shorter lag phases

183-185: These sentences are rationale for your experimental approach. As such, they would fit well towards the end of the Introduction. Similar comments apply to lines 173-175.

185 between lignin-amended and unamended conditions. It would be helpful to specify here that the proteomic data were obtained from pellet samples.

187 The analysis detected a total ………. (Otherwise it reads as if the analysis affected the experiment.)

200-202: This sentence could be split and re-shaped for ease of reading.

214: “To support this…..” How about alternative wording e.g. Further evidence for complex formation…..

218-220: Worth reshaping this sentence for ease of reading and increased clarity.

325: Is the data set in Fig 4 amenable to 3 way ANOVA analysis? That could be used to support the statements about catecholate levels.

356-7: BS08 is a member of the order Alteromonadales closely related to the gill-resident symbiont Teredinibacter turneri. It is also gill-resident rather than gut resident (Altamira et al 2014 Mol Ecol 23: 1418-1432)

Fig 1. It would be easier to make comparisons if the X and Y axes were over the same scale in left and right graphs. The pdf of this figure does not show “early exponential” and “late exponential” labels for the left and right images. The legend needs to specify that upregulation in lignin-amended relative to unamended. Also, it should specify that the proteomics were performed on pellets from the culture.

Fig 2. Y axis for 2A is missing in the pdf. Are error bars possible for this figure? Useful to specify pellet samples.

Fig. 3. With smaller data point markers, it would be possible to see error bars better. Should state mean ± SE, n= . Perhaps the actual data points could be shown as well?

Fig. 4 Should state mean ± SE, n=

Fig. 5 Suggest resequencing wording:- SDS-PAGE of secretome of T. lignolytica BRL6-1 cultured in lignin-amended and unamended conditions.

6. PLOS authors have the option to publish the peer review history of their article (what does this mean?). If published, this will include your full peer review and any attached files.

Reviewer #1: No

Reviewer #2: No

---

## [Author Response · Author response to Decision Letter 0]

23 Aug 2020

Response to Reviewers

This document is also included as an attachment, in which reviewer comments are in black, and our response is in blue. 

Reviewer #1: 

Manuscript appears to be of significant quality and relatively well written. Using “omics” and complementary biochemical studies the manuscript paints a wonderful picture of what appears to be a novel bacterial anoxic lignin utilization system. Although nice, everything is circumstantially connected and still requires the difficult task of detailed biochemical analysis of the proteins involved. As such, the title comes off as overselling what has actually been done. I suggest retitling.

We have retitled the manuscript, which is now: “Lignin Induced Iron Reduction by Novel sp., Tolumonas lignolytic BRL6-1”.

Also, I think that the methods need more detail.

We have expanded the details in the methods section.

Also, while the presentation is nice, I think the results shown are limited. When I look at the SDS-PAGE I ask what of the 11kDa and 8kDa bands? If just artifacts there should be a comment. These were >10kDa fraction correct? So what are these bands?

These bands are artifacts – likely from the lignin phenolics interacting with the proteins that were not removed during the TCA precipitation. Because these data were obtained using a 10 kDa cutoff filter, these bands were considered to be too close to the cutoff, and therefore cannot be further investigated with confidence. This is now addressed in the manuscript.

Almost all figures would benefit by more contrasting colors selection blue/light blue or black/blue. What about orange/blue or red/green? This included figure 4. The patterns should be replaced with colors.

Colors have been changed to give more contrast between results, and all figures revised. 

Abstract:

Line 27: “10kDa fraction” is not important information for the abstract.

This has now been addressed by removing this information from the abstract. 

Introduction:

Last paragraph: Does not really serve the purpose of the typical closing paragraph of an “Introduction”. ‘In this work we…..’ is missing. Instead a hypothesis is introduced, which seems to be the working hypothesis in consideration of the proposed research results.

This has now been addressed by rewriting this paragraph. In addition, some of Reviewer #2’s suggestions have also been incorporated into this paragraph. 

Materials and Methods:

General note: Bits of the Materials and Methods section (often the first sentence of a paragraph) out of place and read more like the Results section.

This has been addressed by rewriting the first sentences of paragraphs in the Materials and Methods section.

Line 109: change “for” to “of”. 

This has now been addressed. 

Proteomics section, Lines 109-133: The section appears to be explain everything twice. Once for the cell pellet and then again for the spent growth media. Perhaps this can be consolidated. I believe data processing was likely done the same in both cases. Consider merging the two paragraphs into one.

We have rewritten parts of these two paragraphs to better highlight the differences in sample preparation. 

Line 112: Remove “Briefly described”. 

This has now been addressed. 

Line 113: Could more information or a reference be included regarding the method for whole cell protein preparation?

This has now been addressed with additional information now provided on cell protein preparation and analysis. 

Line 114: “…searched using MS-GF+….” 

This has now been addressed.

Line 115: spelling of “Porcine”. 

This has now been addressed.

Line 125-126: description of 10kDa filtration is not accurately described. I believe this was a molecular or ultrafiltration centrifugal filter device similar to that described on Line 148. Amicon?

The reviewer is correct that ultrafiltration with an Amicon filter was used. We have rewritten this sentence to include this information. 

Line 143: Delete “In order to confirm that iron was bound to lignin,” 

This has now been addressed.

Line 149-151: All volumes provided for composition, except for the 500 mM buffer.

This has now been addressed – the 60 μL of 50 mg L-1 ferrozine was dissolved in the 500 mM potassium acetate buffer. We have reworded the sentence for clarity. 

Line 156: Delete “To determine if BRL6-1 is producing catechol-like siderophores known as catecholates” and replace with something smaller such as “To quantify catecholates …..”.

This has now been addressed - we have reworded the sentence for clarity.

Line 157: “….as previously described.” 

This has now been addressed.

Results and Discussion

Line 175: The “…shorter lag phase….” I think needs more discussion. This is a theme throughout manuscript and I’m not sure that it has been hashed out enough. It seems clear lignin is doing something. It does not seem to be a direct Fe effect based on the results. Instead I believe the authors are pointing to a hypothetical protein of about 20kDa which is actually not directly connected to the other results. Circumstantially perhaps. So then it must the carbon. Then that would mean that the bug is more efficient in (or regulated for preference) in the use of lignin degradation products as compared to glucose.

We have added a sentence explicitly addressing the shorter lag phase: “The shortening of the lag phase is expected for diauxic microbes as an adaptative trait for frequently changing environments.” We also have revised this section as well as the “Lignin-amended cultures shift in carbon metabolism and energy production” section to address these questions with more clarity. This includes a metabolomics analysis that was collected in parallel with the cellular protein analysis.

Line 207: “For our protein….” Should be stated better. Maybe, “This CBM protein….”

This has now been addressed - we have reworded the sentence for clarity.

Line 227: Figure Legend, KDPG is missing the enzyme descriptor “aldolase”. 

This has now been addressed.

Line 237-239: Not complete thought. 

This has now been addressed - we have added more content to the sentence.

Line 260: Line 257 has PEPCK spelled out upon first use in text, but KDPG is not spelled out and I believe it is the first use as well. 

This has now been addressed - we have spelled out KDPG.

Line 262: “dynamic” seems like a strange word. Should it be maybe “unusual”?

This has now been addressed and removed with the restructuring of the section.

Line 269-271: So either way justifies the same outcome?????? That does not sound scientific!

We have made revisions to this section of the manuscript for clarity and data interpretation. This includes metabolomic data that was completed in parallel with the cellular proteomic collection to help in our interpretations.

Line 292-293: This sentence and surrounding are not communicated well. Please make more clear. 

This has now been addressed - we have reworded the paragraph for clarity.

Line 312: Fe(II/III) should just be written out. 

This has now been addressed.

Line 343: “Differential banding was observed at 20kDa….” 

This has now been addressed.

Line 356: “…homology to a hypothetical…” 

This has now been addressed.

Line 383: Yes, the manuscript definitely illustrates potential of a molecular mechanism. There appears to be a leap at the end to connect all the dots in story. Do not feel as though it is all substantiated. Perhaps a new title and trimmed message may help bridle the wild story!

We hope with the current edits that the story is more direct.

Line 511: Issue with reference.

This has now been addressed.

Reviewer #2: 

Where the response given to editorial questions is “no”, this is because the system does not permit a more nuanced response such as “probably with further information”, “needs clarification”, “not yet” or “not entirely”. These would have been my choices otherwise.

Main comments

This is an interesting study of bacterial lignin degradation capabilities. Whereas fungal mechanisms of lignin degradation have become quite well defined, bacterial processes particularly in anaerobic conditions are much less well understood. This study provides some welcome new insights into bacterial mechanisms, with hypotheses suggested for further exploration. Some broader ecological context in the Introduction might be useful too, as the final mineralisation of lignocellulosic substrates typically occurs in soils or sediments. As these mechanisms are of interest to researchers such as ecologists interested in carbon fluxes and food webs, who are less familiar with microbiological methodology, it would be helpful if the text were made more accessible to non-specialists. Consistently specifying where data derived from pellet or supernatant are being presented would be helpful. Some of the detailed comments below give examples where adjustments could assist in this respect. Also, a

sentence or two in the conclusions about the ecological implications of the findings would be useful.

We have added a few sentences to tie in the ecological context. However, we want to keep the main focus on the physiology of the organism since the goal of this project is to give more insight into anaerobic lignin degrading bacteria, such as BRL6-1. 

Using biological replicates of proteomic data is a particularly valuable feature of this study. To properly evaluate the statistical treatment of the findings, clarification of details of experimental design in the methods, with linking wording in the Results and figure legends is needed. Some additional analysis may also be worthwhile. A flow diagram of the experimental design showing levels and type (technical and biological) of replication, type of sample (supernatant, pellet) and experiments undertaken would assist the reader. The text would need to be adjusted to link to wording used in this diagram.

We have reworded portions of the manuscript to better link the methods with the analyses and results. This also includes the figure legends. Due to four different growth curves being described in the methods, having another element such as a figure with 4 different flow diagrams may complicate it more for the reader. We feel that the other adjusts made based on your recommendations mitigates the need for the flow diagrams. 

Detailed comments and suggestions of edits. Numbers refer to the line numbers in the left margin

24: develops a higher biomass? 

This has now been addressed. We changed the word to “accumulates”. 

66: Mass production……is not possible. 

This has now been addressed.

70-71: Restructure sentence for increased clarity. 

This has now been addressed - we have reworded the sentence for clarity.

83: worth mentioning that this bacterium was isolated from forest soil (and possibly an anoxic layer within the soil) this point could be briefly addressed in the Conclusions too. 

This has now been addressed. We added a descriptor in this sentence to mention that it is a soil bacterium. We also added a statement that this organism experiences frequent strong fluctuations in redox in the tropical soil it was originally isolated from, so innate capability for iron reduction fits with the ecology of this system.

84: …… was demonstrated to have. 

This has now been addressed.

87: perhaps “BRL6-1 secretes a protein……”? 

This has now been addressed. Please note that the paragraph itself has been modified based on both Reviewer #1 and #2’s suggestions.

99: Perhaps “Triplicate cultures were grown anaerobically at 30°C and triplicate uninoculated…” Wording needs to be more specific about replication, 3 cultures per experimental condition presumably.

This has now been addressed. We have reworded the sentence for clarity: “For each experimental condition, cultures grew at 30°C anaerobically (n=3). Uninoculated bottles served as abiotic controls (n=3, unless stated otherwise).”

124: 100mL samples from the cultures? 

We have reworded this sentence for clarity as well as surrounding sentences.

136: Suggest giving a bit more detail about what “spectral counts” means. 

We have added a brief definition here for spectral counts and what they represent. 

144: as described by Jeitner 

This has now been addressed.

156: To determine whether BRL6-1 produces….. 

This has now been addressed and the sentence has been reworded.

157: as described by…… or as previously described 

This has now been addressed.

161: absorbance was read 

This has now been addressed.

161-162: “Samples……..in triplicate.” Does this mean that triplicate technical replicates were taken from each of three cultures and three abiotic controls, or one biological replicate was taken from each replicate culture and each abiotic control?

We have rephrased the sentence to the following: “Triplicate technical replicates were taken from each of the three biological replicates as well abiotic controls for each condition in duplicate.”

175: a shorter lag phase or shorter lag phases 

This has now been addressed.

183-185: These sentences are rationale for your experimental approach. As such, they would fit well towards the end of the Introduction. Similar comments apply to lines 173-175. 185 between lignin-amended and unamended conditions. It would be helpful to specify here that the proteomic data were obtained from pellet samples.

We have added this detail to introduction. In addition, we have specified that the proteomic data was obtained from the cell pellet samples. We kept lines 183-185 as well as 173-175 as a reminder of our rationale now mentioned in the introduction.

187 The analysis detected a total ………. (Otherwise it reads as if the analysis affected the experiment.)

This has now been addressed.

200-202: This sentence could be split and re-shaped for ease of reading.

This has now been addressed. We have reworded the sentence for clarity.

214: “To support this…..” How about alternative wording e.g. Further evidence for complex formation…..

This has now been addressed. We have reworded the sentence for clarity.

218-220: Worth reshaping this sentence for ease of reading and increased clarity.

This has now been addressed. We have reworded the sentence for clarity.

325: Is the data set in Fig 4 amenable to 3 way ANOVA analysis? That could be used to support the statements about catecholate levels.

Since the lignin un-amended cultures did not contain detectable catecholates, we completed a two-way ANOVA comparing lignin amended cultures vs abiotic controls across time. This is now described in the revised manuscript: “Two-way ANOVA analysis supported this finding with no significant effect between the cultures and abiotic controls (p = 0.159), growth phases (p = 0.851), or their interaction (p = 0.308).” Also this analysis is mentioned in the methods: “Arnow results of the lignin condition between cultures and abiotic controls across time were analyzed with a two-way ANOVA using the stats package in R.”

356-7: BS08 is a member of the order Alteromonadales closely related to the gill-resident symbiont Teredinibacter turneri. It is also gill-resident rather than gut resident (Altamira et al 2014 Mol Ecol 23: 1418-1432)

We appreciate the author taking the time to check references and catch this mistake on our part. We have removed “gut” and explained the role of digestion of the gill endosymbionts for the shipworm and why this is related to our results for Tolumonas lignolytica BRL6-1.

Fig 1. It would be easier to make comparisons if the X and Y axes were over the same scale in left and right graphs. The pdf of this figure does not show “early exponential” and “late exponential” labels for the left and right images. The legend needs to specify that upregulation in lignin-amended relative to unamended. Also, it should specify that the proteomics were performed on pellets from the culture.

The following has been addressed: (1) labels were added for each phase, (2) the caption of the figure specifies that all protein abundances of the lignin amended cultures are relative to the unamended controls, and (3) the caption specifies that this proteomic analysis is from the cell pellet fraction (“cellular protein abundance"). Due to the restraints of the analysis output (edgeR package in R), the X and Y axes cannot be over the same scale. 

Fig 2. Y axis for 2A is missing in the pdf. Are error bars possible for this figure? Useful to specify pellet samples.

The Y axis was added to the figure. For edgeR analysis, the log fold change accounts for the triplicate samples but gives one final output value- which means error bars cannot be possible. We have now specified pellet samples in the caption (“cellular proteins”). 

Fig. 3. With smaller data point markers, it would be possible to see error bars better. Should state mean ± SE, n= . Perhaps the actual data points could be shown as well?

The data point markers have been reduced in size from 6 to 3. We have indicated that the error bars represent standard deviation (not labelled prior), not standard error. “n” has now been added to the caption to indicate n=3 for each treatment. 

Fig. 4 Should state mean ± SE, n=

We have indicated that the error bars represent standard deviation (not labelled prior), not standard error. “n” have been stated in the caption. 

Fig. 5 Suggest resequencing wording:- SDS-PAGE of secretome of T. lignolytica BRL6-1 cultured in lignin-amended and unamended conditions.

This has now been addressed with the title reworded.

---

## [Editor Report · Decision Letter 1]

27 Aug 2020

Lignin Induced Iron Reduction by Novel sp., Tolumonas lignolytic BRL6-1

PONE-D-20-14077R1

Dear Dr. DeAngelis,

We’re pleased to inform you that your manuscript has been judged scientifically suitable for publication and will be formally accepted for publication once it meets all outstanding technical requirements.

Kind regards,

Daniel Cullen

Academic Editor

PLOS ONE
---

## [Editor Report · Acceptance letter]

7 Sep 2020

PONE-D-20-14077R1 

Lignin Induced Iron Reduction by Novel sp., *Tolumonas lignolytic* BRL6-1 

Dear Dr. DeAngelis:

I'm pleased to inform you that your manuscript has been deemed suitable for publication in PLOS ONE. Congratulations! Your manuscript is now with our production department. 

Kind regards, 

on behalf of

Dr. Daniel Cullen 

Academic Editor

PLOS ONE